# Potential Applications of Frass Derived from Black Soldier Fly Larvae Treatment of Food Waste: A Review

**DOI:** 10.3390/foods11172664

**Published:** 2022-09-01

**Authors:** Noor Ezlin Ahmad Basri, Nur Asyiqin Azman, Irfana Kabir Ahmad, Fatihah Suja, Nurul Ain Abdul Jalil, Nur Fardilla Amrul

**Affiliations:** 1Department of Civil Engineering, Faculty of Engineering and Built Environment, Universiti Kebangsaan Malaysia (UKM), Bangi 43600, Selangor, Malaysia; 2Sustainable Urban Transport Research Centre (SUTRA), Faculty of Engineering and Built Environment, Universiti Kebangsaan Malaysia (UKM), Bangi 43600, Selangor, Malaysia; 3Department of Earth Science and Environment, Faculty of Science and Technology, Universiti Kebangsaan Malaysia (UKM), Bangi 43600, Selangor, Malaysia

**Keywords:** black soldier fly larvae frass, frass properties, frass applications, pre-treatment of food waste, post-treatment of frass, frass composting

## Abstract

The disposal of large amounts of food waste has caused serious environmental pollution and financial losses globally. Compared to alternative disposal methods (landfills, incineration, and anaerobic digestion), composting by black soldier fly larvae (BSFL) is a promising alternative for food waste management. Despite extensive research into larval biomass, another valuable by-product generated from BSFL composting is BSFL frass. However, limited information is available for its potential application. The applications of BSFL frass can be intensified by understanding its physicochemical characteristics, benefits, and challenges of BSFL frass derived from food waste. BSFL frass is harvested after 9–23 days of the experiment, depending on the substrate used in the composting process. The generated BSFL frass could exceed 33% of the original weight of the substrate. The physicochemical characteristics of BSFL frass are as follows: the temperature after harvest is 24 °C to 27 °C, pH is 5.6–8.0, moisture content is 30 to 72%, C/N ratio is 8:1 to 27:1, high nitrogen, phosphorus, and potassium (NPK) content, and low heavy metal content. This paper reviews the characteristics, benefits, and application of BSFL frass. It will also investigate the challenges of using food waste substrates to produce BSFL frass, as well as the best way to pre-treat the food waste substrate and post-treat the BSFL frass.

## 1. Introduction

Solid waste generation is expected to rise due to rapid population growth and urbanisation. The world generates 1.3 billion tonnes of solid waste per year [1]. The amount of waste is projected to increase from two billion tonnes in 2016 to 3.4 billion tonnes in 2050, with food waste accounting for over half of all solid waste generated globally [2]. Food waste is a major environmental and economic burden, as well as one of the most complex aspects of solid waste management. It is feasible, however, to create valuable products by recovering energy and nutrients from food waste [3,4]. One of the most innovative waste management strategies for recycling food waste is to feed the food waste to insect larvae. The composting of food waste by black soldier fly larvae (BSFL) has attracted considerable interest among researchers, the media, the public, and waste management entrepreneurs since it was first introduced in the 1990s [5]. The life cycle of BSFL consists of five stages (Figure 1), the eggs, larvae, prepupae, pupae, and adults. The life span of the adult fly is short, and the larval and pupal stages make up most of the BSF life cycle while the adult flies are harmless to humans because they do not have stingers, and their mouth parts or digestive organs only allow them to drink water [6]. The BSFL composting is simple, requires a small space and is cost-effective because the process typically takes place in boxes, bins, or containers; the BSF larvae are fed once or periodically [7]. The BSFL self-harvesting strategy also helps to reduce highly technical operational skills, which is beneficial to the circular economy and provides a sustainable operation for low- and middle-income countries [7,8]. Considering the large volume of organic waste generated daily, composting organic waste using BSFL is strongly recommended for all regions, including residential areas and universities [9]. BSFL composting, therefore, is a sustainable and environmentally friendly technique to reduce waste. According to Izzati et al. [10], when introducing new technology or techniques into any system, the sustainability and environmental impact must be considered.

BSFL can feed and grow on a broad range of decaying plant and animal matter, including manure, food waste, municipal waste, and rotting plant material [11]. They can eliminate more than 50% of total organic waste in a short period (two weeks) under ideal conditions [12,13]. BSFL is unique in that it can convert organic wastes into protein and fat-rich biomass that is safe for animal consumption and, thus, is suitable as a substitute for the standard proteins in animal diets. The nutritional content of the larvae, about 40% larval protein and 30% larval fat, is beneficial for producing goods, such as animal feed and biodiesel [14]. In the emerging industry of insects for food and animal feed, the second by-product generated in a large quantity in BSFL composting is frass. BSFL frass is a compost-like material generated by feeding low-quality organics to BSFL living in high larvae density with bacteria. BSFL frass could recapture nitrogen and phosphorus from the already existing food chain to be reused as fertiliser and reduce the demand for chemical fertilisers [15]. Potential revenue loss may come from BSFL frass because of high potential in producing high-value products, such as fertiliser, biochar, and fish feedstock. To avoid revenue losses on BSFL frass, studies on the characteristics of BSFL frass are needed [3]. It is essential to determine the benefits and challenges of using BSFL frass [15]. Not much is known about the challenges in using food waste as a substrate to produce BSFL frass and the post-treatment of BSFL frass. Therefore, this paper aims to determine the characteristics, benefits, and applications of BSFL frass. It will also investigate the challenges of using food waste substrate to produce BSFL frass, the suitable pre-treatment of the food waste substrate and the post-treatment of BSFL frass.

## 2. Methodology

This review paper uses the secondary data from online database tools and scientific domains, such as Science Direct, Google Scholar, Research Gate, Web of Science, and Scopus (from 2002 up to August 2022). The review uses keywords to compile the published literature on BSFL frass and sub-study themes about BSFL frass production from food waste substrate. In summary, this study gathers all available information on frass from BSFL treatment and describes the pre-treatment of food waste and post-treatment of frass.

## 3. Black Soldier Fly Larvae Frass

Researchers use the term residue for the BSFL second by-product, which has the same meaning as the frass. Diener et al. [12] stated that the second by-product of BSFL treatment is residue or BSFL digestate; while, according to Green and Popa [16], Quilliam et al. [17], and Zahn [18], frass is the waste excreted by the larvae of BSFL composting. Frass is a compost-like material [19] and has the characteristics of immature compost [20,21,22]. In a commercial context, BSFL frass often refers to a mixture of primarily BSFL faeces, substrate residues, and shed BSFL exoskeletons [23]. Schmitt and de Vries [15] contended that frass is a mixture of uneaten feed materials, insect derivatives, such as skins and faeces, and a microbial population that carries out fermentation.

### 3.1. Harvesting BSFL Frass

BSFL frass is usually harvested after 9–23 days of BSFL composting, depending on the type of waste used [3,11,20,23,24]. Alternatively, the frass is harvested when more than 40–90% of larvae have transitioned to the prepupae stage [23,25,26]. The harvest time is crucial in full-scale BSFL treatment operations to ensure maximum larval production and good quality product [7]. Harvested BSFL frass may come in dry or wet compost. Dry BSFL frass is produced mainly as granular dark brown granular texture and has lower water content [21]. Meanwhile, wet BSFL frass is dense, grey in colour, has high moisture content, and has the consistency of thick, moist clay [27]. The good BSFL frass texture will allow easy storage, packaging and transport without further transformation or stabilization [13].

### 3.2. BSFL Frass Weight

BSFL was introduced as an efficient way to convert biowaste into larvae products and frass while reducing a large volume of waste [28]. One thousand kilograms of food and vegetable substrate in the BSFL treatment process can yield 25.0% (250 kg) BSFL frass and 12.5% (125 kg) BSFL [29]. Salomone et al. [13] reported that BSFL treatment of 30 tonnes of food waste per day generated 33.3% (9990 kg/day) BSFL frass and 7.7% (2310 kg/day) BSFL. Another study by Quilliam et al. [17] found that the bioconversion of brewery waste produced 33.8% (738 kg) BSFL frass and 2.5% (55.1 kg) BSFL from 2182 kg of the substrate, while 858 kg of poultry manure produced 62.9% (540 kg) BSFL frass and 6.4% (54.9 kg) BSFL. In summary, the weight of BSFL frass could exceed 33% of the original substrate weight and the weight generated will be affected by the types of substrates.

## 4. Physiochemical Characteristics of BSFL Frass

### 4.1. Temperature

The harvested BSFL frass has a temperature of 24 °C to 27 °C [20,30]. The rapid composting of BSFL occurs in the mesophilic phase; BSFL continues to transform the substrate, while higher air circulation during the BSFL composting process maintains a relatively constant waste temperature (about 30 °C) for optimum waste consumption by BSFL [31]. Rearing BSFL at the optimum temperature improves their ability to reduce *Escherichia coli* [32].

### 4.2. The pH of the BSFL Frass

Referring to Table 1, BSFL frass from food waste range from the lowest of 5.6 in fruit and vegetables and the highest as 8.0 pH value in mixture of food waste, chicken faeces, and sawdust (3:2:1 ratio); and maize straw substrates. The pH of BSFL frass typically ranges between 7.0 and 8.0, which is the optimum range for promoting plant growth [33] and providing a conducive environment for the beneficial bacterial communities in BSFL frass [34].

### 4.3. Moisture Content of the BSFL Frass

Referring to Table 1, the driest BSFL frass can be seen from brewery spent grain at 30% moisture content, and the wettest at 72% moisture content from the food waste, chicken faeces, and sawdust mixture (3:2:1 ratio). The high moisture content of frass makes harvesting difficult because the frass has a clay-like texture, making it hard to collect the BSFL biomass under the wet frass. One way to harvest the BSFL biomass from the wet frass is by rinsing it under running water [39]. However, this method washes out the frass, which is a waste since the frass can be processed into beneficial products for agricultural use. Using a suitable pre-treated waste in BSFL treatment (processing parameters and feeding strategies) can produce dry frass. It is worth noting that dry frass is considered immature compost since the organic waste must undergo rapid composting for about two weeks which may contains phytotoxins compounds that may inhibit plant growth. Many researchers recommend post-treatment of the dry frass to ensure its maturity and stability [40].

### 4.4. The C/N Ratio of the BSFL Frass

There are numerous approaches for determining organic fertiliser stability and behaviour in soil, and the soil C/N ratio may reveal vital soil information. As shown in Table 1, C/N ratio of BSFL frass derived from different types of food may range from 8:1 to 27:1; kitchen waste range from 8:1 to 17:1, municipal food waste range from 8:1 to 9:1, household food waste at 17:1, fruits and vegetables at 27:1, okara and wheat bran at 8:1 (fresh frass), okara and wheat bran (composted frass) at 10:1, and brewery spent grain at 17:1. In a well-conducted composting process, the C/N will decrease constantly. This is due to the biological mineralization of carbon compounds and loss as CO_2_ [41].

### 4.5. Nutrient Content (NPK) of the BSFL Frass

Most insect frass has a high nutrient content, such as organic N and P [18]. Chen et al. [42] stated that the continuous movement of BSFL could reduce the BSFL frass temperature, which helps retain nitrogen in the BSFL frass and ensure a high nitrogen content in the BSFL frass. The nutrient content of BSFL frass is influenced primarily by the nutrient content of the substrate fed to the BSFL [33]. Referring to Table 1, BSFL frass derived from food waste have total nitrogen content range from 0.6 to 4.8, total phosphorus content ranges from 0.8 to 2.5 and total potassium range from 0.2 to 2.1.

As shown in Table 2, for agronomic purposes, total nitrogen, total phosphorus, and total potassium are acceptable at the range of greater than 0.6, 0.22, and 0.25, respectively. Commercial organic fertilizer has 2.3% total nitrogen content, 2.3% total phosphorus content, and 2.3% total potassium content. Commercial chemical fertiliser, however, is composed of higher nutrient content: 16% total nitrogen, 16% total phosphorus, and 16% total potassium. If we compare this to the BSFL frass nutrient content in Table 1, fresh frass and composted frass from okara and wheat bran substrates have higher total nitrogen content (3.2% and 4.8%) than the commercial organic fertilizer (2.3%). Total nitrogen of BSFL frass from household food waste (2.2%); mixture of food waste, chicken faeces, and sawdust (1.7%); and fruit and vegetables (1.8%) are also close to the value of commercial organic fertilizer. However, total phosphorus and total potassium composition of BSFL frass from household food waste have not passed the preferable range. The lowest total nitrogen (0.6%) of BSFL frass can be seen in maize straw substrates, however, it also has total phosphorus (2.5%) and total potassium (2.1%) content like the commercial organic fertilizer. It is possible to observe that the mixture of two or three types of food waste present the acceptable and consistent value of BSFL frass nutrient content. Food waste, chicken faeces, and sawdust (3:2:1 ratio) has 1.7% (total nitrogen), 1.1% (total phosphorus), and 2.1% (total potassium). Respectively, okara and wheat bran mixture have 3.2–48% (total nitrogen), 0.8% (total phosphorus), and 0.5% (total potassium). Therefore, the mixture of two or different types of substrates may be preferable to produce high nutrient BSFL frass.

### 4.6. Nutrient Composition of Food Waste Attribute to the BSFL Frass Composition

Referring to the study of Palma et al. [50], seven almond by-product (hulls and shells) samples were obtained from California processors, and BSFL frass samples were evaluated for their composition in terms of ammonium, phosphorus, and potassium production. Using the results shown in Table 3, the highest protein (67.7 g kg^−1^) and calcium (2.8 g kg^−1^) was obtained from sample 5 has contributed to the lowest ammonium (400 ppm) production in BSFL frass and highest phosphorus concentration (825.0 ppm). The lowest protein (40.1 g kg^−1^), highest fat (31.0 g kg^−1^), and highest starch content (5.33 g kg^−1^) obtained from sample 1 was attributed to the considerable amount of ammonium (615.0 ppm), phosphorus (355.0 ppm), and potassium (35.4 ppm). Sample 7 has the lowest fat (14.6 g kg^−1^), lowest calcium (1.9 g kg^−1^), and lowest sugar (53.2 g kg^−1^) and, as a result, has the lowest potassium content (17.9 g kg^−1^). Sample 4 has the highest amount of sugar (291.3 g kg^−1^) and has the highest amount of potassium in BSFL frass (44.6 g kg^−1^). The highest amount of potassium can also be found in sample 3 (44.6 g kg^−1^). However, the highest ammonium content (5695.9 ppm) and lowest phosphorus content (227.5 ppm) can be found in sample 2, where the sample has the lowest starch (3.57 g kg^−1^). The results from this study have shown that high protein and calcium in food waste composition will cause a good effect in the BSFL frass composition. The lowest starch composition in food waste, however, has resulted in the highest ammonium and lowest phosphorus content in BSFL frass. Sugar, however, may be attributed to the amount of potassium in the BSFL frass.

### 4.7. Heavy Metals in BSFL Frass

BSFL is one of the microorganisms which biologically accumulate heavy metals in their tissues, leaving their frass with low concentration of heavy metals. Previous studies show that large quantities of mercury have been added to the BSFL feedstock to be observed in a 13-day experiment and resulting in low mercury levels in the BSFL frass and were noted to be below the European Union’s (EU’s) threshold values of 0.7–10 mg Hg/kg [20]. Assessment of the heavy metal contents in BSFL frass showed considerable removal of As, Cd, Pb, and Fe [30]. The result showed 92–98% (0.0002–0.0008 mg/kg) removal of As, 99% to 100% (0.00029–0.00170 mg/kg) of Cd, and 80% to 90% (0.001–0.002 mg/kg) of Pb. Although Fe in BSFL frass content has a removal of 31% to 69% (1.70–2.56 mg/kg) from the initial substrate and Ni has shown no observable removal (0.01–0.03 mg/kg) in the experiment, both values still comply with maximum allowable level for heavy metals in organic fertilizer [30]. Salomone et al. [13] has also measured the concentrations of toxic and essential metals in the BSFL frass fed with food waste substrates and found that the concentration of toxic and essential metals was below the limits stated in the Italian regulation for fertiliser. BSFL frass has low concentrations of heavy metals because of the ability of BSFL to reduce and accumulate various forms of heavy metals in the BSFL treatment process. However, there is concern about heavy metals accumulated in BSFL biomass that may contains cadmium, lead, mercury, zinc, and arsenic from the biowaste, and the amount of the heavy metals absorbed might exceed the maximum allowable amounts of the regulations on animal feed [7].

### 4.8. Maturity and Stability of BSFL Frass

Early studies on BSFL frass have always focused on only NPK and micronutrients content to analyse BSFL frass as organic fertiliser. However, upon testing BSFL frass on several types of plants, stunted plant growth and reduced biomass production in the plants was observed. Therefore, there is concern on checking the maturity and stability of BSFL frass to identify the possible application. Table 1 shows a compost level preferred for agronomic purposes. According to Diaz et al. [41], to mimic fertiliser analysis, compost analysis has concentrated on NPK and micronutrient contents while compost quality and maturity are controlled using chemical parameters, such as pH, ammonia, C/N, as well as plant growth, germination tests, and microbial tests.

Within a short period of BSFL rapid composting (two weeks to a month), organic wastes fed by the BSFL may not be properly composted [36,37]. The BSFL composting process must also stop when the larvae reach the prepupae stage, as a result, producing impartial compost and immature compost. Referring to Diaz et al. [41], maturity of a compost can be referred to a compost’s level of phytotoxicity. Immature compost basically has a high level of phytotoxicity and will tend to have more growth inhibitors for plants than a mature compost. Phytotoxins, such as heavy metals, phenolic components, organic acids, and salt accumulation, are common in immature compost [51]. Meanwhile, the stability of a compost can be identified when one that is no longer undergoing rapid decomposition and whose nutrients are tightly bonded; unstable compost, on the other hand, may either release nutrients into the soil owing to additional decomposition or tie up nitrogen from the soil [41].

Temperature is an essential factor in determining whether decomposition proceeds at the mesophilic or thermophilic level, or even reaches the maturity level to generate natural plant fertiliser [43]. According to Table 1, BSFL frass temperature range from 24 °C to 27 °C, if compared to the temperature of other compost, as shown in Table 2, compost from windrow composting (26–28 °C), composting bin (30 °C), and aerated composting (32 °C); all reach an ambient temperature and is considered to have entered maturation phase [44,47]. As suggested by Cooperband [52], the optimum temperatures for bacterial decomposition are at 21–49 °C.

The pH value of compost is worth measuring since it can be used to track the decomposition process. At the beginning of the composting process, the pH value of compost, which indicates acid content, is a result of accumulated acid generation and acid decomposition to produce CO_2_ and heat. The acidic condition is caused by the microbial breakdown of organic materials and the generation of organic acids. The pH value of compost will increase over time as the acids were consumed. The pH rise can also be explained by ammonia generation via ammonification and organic nitrogen mineralization via microbial activity. The mineralization of nitrogen molecules, such as nitrates, nitrites, and other organic acids is also involved in the process [44]. According to Table 1, the pH of BSFL frass from food waste ranges from 5.6 to 8.0 and Table 2 shows the pH value of compost from windrow composting (6.8), compost bin (7.5), aerated composting (6.5–7.5), and digestate from anaerobic digestate (8.1) are all close to the pH value of mature compost 6–8 [48].

Diaz et al. [41] recommend that at the end of the composting process, the water content should be quite low (about 30%) to prevent any further biological activity in the stabilized material. Although, from a production perspective, the moisture level of finished composts should be between 35% and 45%. If the moisture level falls below 35%, a lot of dust will be created when these composts are transported or spread to fields [49]. In Table 1, moisture content of BSFL frass from kitchen waste ranges from 50 to 63%, municipal food waste ranges from 63 to 65%, household food waste at 56%, and mixture of food waste, chicken faeces, and sawdust (3:2:1 ratio) at 72%, fruit and vegetables at 10%, maize straw at 38% and brewery spent grain at 30%. BSFL frass ranges more than 45% from kitchen waste, municipal food waste, household food waste and mixture of food waste, chicken faeces, and sawdust (3:2:1 ratio) in anaerobic conditions showing the condition of BSFL frass is not mature enough. BSFL frass derived from fruit and vegetables is at 10% which is not suitable for agronomic purposes and may lead to hydrophobicity and be difficult to rewet. BSFL frass from maize straw and brewery-spent grain are in the acceptable range for mature compost showing stabilized characteristics for agronomic purposes. As shown in Table 2, moisture content of compost from aerated composting is around the range for stabilized compost at 32–33%. Compost from windrow composting contains 46–61% moisture and digestate from anaerobic contains the highest moisture (85%) compared to BSFL frass and other composting techniques.

There are numerous approaches for determining organic fertiliser stability and behaviour in soil, and the soil C/N ratio may reveal vital soil information. Compost with a C/N ratio less than 20:1 is beneficial to plants because the organic nitrogen has mineralized to inorganic nitrogen, which is then available for plant absorption. However, compost with a C/N ratio greater than 30:1, is more likely to immobilise nitrogen for plant uptake [30]. Referring to Table 1, all the BSFL frass derived from food waste range from 8:1 to 27:1 which have a C/N value lower than 30:1, proving the stability of BSFL frass in compost. Table 2 shows the C/N ratio for the conventional organic waste treatment methods, where the C/N ratio of the compost from windrow composting, digestate from the anaerobic digester, and BSFL frass are within the preferred limit for agronomic purposes, such as fertiliser and soil amendment. However, the C/N ratio for the compost from the aerated composting technique is greater than 30:1 [45], which may cause nitrogen immobilisation. Therefore, it is prudent to evaluate the waste characteristics for C/N ratio and load the waste into the reactor to maintain the required C/N ratio for good quality compost.

Although the C/N ratio can provide some information into the stability and behaviour of compost, germination tests on plant growth can provide more detailed performance data. Song et al. [37], has stated that BSFL rapid composting is insufficient for removing phytotoxins from the okara and wheat bran waste because a seed germination test has indicated that the presence of phytotoxins, and Fourier-transform infrared spectroscopy (FTIR) examination revealed that fresh harvested frass had the greatest phenol compounds compared the other two types of BSFL frass which has undergo post-treatment in aerated composting and natural composting. Kawasaki et al. [36] has also claimed that BSFL frass could be an incomplete compost due to its chemical composition and microbial test. Higher ammonium nitrogen and lower nitrate nitrogen was present in BSFL frass derived from household food waste. This significant amount of nitrogen is typical of biologically unstable compost and has the same characteristics as poultry manure compost. A microbial test to analyse plant pathogens in BSFL frass in the household food waste has demonstrated the high relative abundance of Xanthomonadaceae, which contains a genus that causes disease in plants. Alattar et al. [27] has published one of the first studies on the stability of BSFL frass derived from kitchen waste as a soil amendment. The study involved analysing maize growth for 10 weeks and resulted in stunted maize growth, which the authors have ascribed to the high concentration of ammonia and presence of phytotoxins in BSFL frass, despite no analysis of phytotoxicity compounds being done.

In summary, BSFL frass derived from food waste may comply with the preferable range of mature compost suitable for agronomic purposes: pH (6–8), ambient temperature, and C/N ratio lower than 30:1. However, BSFL frass derived from food waste mostly have higher than 45% moisture content which greatly affect to the immaturity of compost. In depth analysis, such as maturity tests, germination tests, and microbial test on BSFL frass is important before applying BSF frass as a soil amendment and using it as an organic fertiliser. Germination tests and microbial tests of BSFL frass derived from food waste have shown that BSFL frass is an immature compost, therefore there is the need for post-treatment of BSFL frass [36,37].

## 5. Benefits of BSFL Frass

According to Bortolini et al. [19], BSFL frass has compost-like properties. The rapid composting of organic waste by BSFL produced compacted BSFL frass with high macronutrients (NPK), micronutrients, and organic matter contents that are readily available for agricultural use [19,20,25,30]. Referring to Table 4, benefits of BSFL frass were listed. BSFL frass has a rich beneficial microbe [53], such as nitrifying and nitrogen-fixing bacteria that make nitrogen available for plant uptake [34]. Nitrogen-fixing and nitrifying bacteria are crucial because fixed nitrogen is a limited nutrient in most ecosystems, and nitrate assimilation into plant roots makes soils more resilient to flood, drought, and land degradation. In addition, by enhancing nitrogen uptake, the high phosphorus concentration in the BSFL frass has aided in promoting nitrogen accumulation in plants since phosphorus is essential for energy transfer [23]. BSFL frass can recapture nitrogen and phosphorus from the food chain for reuse as fertiliser, thus reducing the need for chemical fertilisers. The presence of chitin in BSFL frass also helps promote plant development and trigger plant defences [33]; chitin also induces improvements in the soil microbiome, thus improving soil fertility [15]. According to Choi and Hassanzadeh [34], BSFL frass is the only plant-digestible form of chitin, and when under environmental stress, the chitin naturally produces antimicrobial peptides that serve as a defence barrier. This can happen because of the uniqueness of BSFL that eventually produces an antimicrobial peptide that reduces pathogenic bacteria, such as *Salmonella enterica* and *Escherichia coli*, during the treatment process. A potentially beneficial effect of insect chitin and metabolites is their ability to stimulate ecological processes by reducing the pest pressure effect on plant. Applying even a minimum amount of BSFL frass chitin to plants results in better growth, more flowers and seeds, and attracts more pollinators.

Almost half of the global waste generation is food waste, which 37% of them goes in landfills and 33% of them is disposed in open dumps area [2]. However, this food stream, which contains high concentrations of organic matter, macro- and micronutrients, if not properly disposed of, might constitute harm to the environment. Therefore, in ecological perspectives, BSFL frass production has contributed to the recycling of nutrients from food waste and has prevented the environment from becoming a concern owing to the emission of greenhouse gases and soil/water contamination with poisonous chemicals and nutrients from leachates [31,40]. Utilizing BSFL frass as a value-added product in BSFL farming has also brought greater economic benefits compared to the net revenue of BSFL animal feed alone. According to a study conducted by Beesigamukama et al. [54], employing BSFL frass fertiliser has boosted farmers’ net revenue by 5–15 times compared to BSF farming alone. One megagram (Mg) of dried BSF larvae (USD 900) produces 10–34 Mg of BSFL frass fertiliser, which is worth USD 3000–10,200. Field trials were also used to assess the agronomic efficacy of BSFL frass fertiliser on maize crops. Maize planted on BSFL frass fertilizer-treated plots had a net revenue that was 29–44% more than maize grown on commercial organic fertilizer-treated plots. Furthermore, smallholder insect farmers who use BSFL frass fertiliser directly for maize growing will generate 30–232% more net revenue than farmers who buy identical BSFL frass fertiliser.

## 6. Applications of BSFL Frass

### 6.1. Organic Fertiliser

BSFL frass is rapidly gaining global attention as an organic fertiliser. BSFL frass has shown similar results in performing as organic fertiliser, even by feeding BSFL to different type of substrates: organic municipal solid waste [30]; food waste [3,13,20]; manure [3,21]; brewery spent grains [38]; mixture of poultry, brewery waste, and green market waste [17]; mixture of pig manure, dog food, and human faeces [55]; and fermented maize straw [25]. Studies on the quality of BSFL frass as organic fertiliser revealed a significant increase in NPK concentrations and a considerable reduction in heavy metals to the acceptable levels set by the regulatory authorities [13,20,25,30]. Referring to Figure 2, BSFL frass shows various applications that could bring benefits in many industries. BSFL frass has a promising potential as an organic fertiliser since it shows similar performance to commercial fertilisers in increasing maize production [38]. According to Quilliam et al. [17], BSFL frass as an organic fertiliser has given comparable results as organic fertiliser in the growth of chili pepper compared with standard local farmer practice using chicken manure fertiliser. Furthermore, the production of one kilogram of BSFL frass can reduce global warming since it requires 12.5 kg less CO_2_ than the commercial fertilizer [15].

### 6.2. Soil Amendments

Adding BSFL frass to the soil in agricultural settings is beneficial to the plant or insect ecosystem [15] and has improved the organic matter quality of intensively cultivated soils [19]. BSFL frass can be a source of soil nutrients without affecting soil hygiene [23]. Choi and Hassanzadeh [34], has suggested that regular addition of BSFL frass to the soil will prevent fungal disease pathogens, such as Rhizoctonia, Fusarium, and Pythium. Quilliam et al. [17] also reported that the plots with the addition of BSFL frass have significantly fewer dead plants than the control plots with synthetic NPK fertilisers, where the cowpea crop has suffered severe Fusarium wilt (*Fusarium oxysporum*) outbreak.

### 6.3. Growing Media

Setti et al. [56] proved that BSFL frass is a suitable growing media to promote soilless agriculture because it can replace the commercial peat used in potted plants. Using 80% commercial peat and 20% BSFL frass as growing media has a beneficial effect on crop growth without causing abiotic stress, especially in the increase in total dry weight, increased leaf area and the number of the production has increased up to 20% for potted plants, such as baby leaf lettuce, basil, and tomato when compared to potted plant production by using commercial peat.

### 6.4. Biochar

Researchers have investigated using insect frass pyrolysed to biochar as a bio-adsorbent for wastewater detoxification in industrial environments [23]. Even though frass from BSFL is not known to produce high-efficiency biochar, insect frass from mealworms (*Tenebrio molitor* Linnaeus 1758) feeding on wheat straw showed the best adsorption performance for bio-adsorbents and have the better capacity (1738.6 mg/g) of absorbing malachite green, (a cationic dye which is a highly toxic dye that may be found in wastewater) when compared to frass fed with wheat bran, raw wheat straw, and raw wheat bran [57]. Researchers are starting to turn their attention to producing biochar from insect frass because of the presence of chitosan, which might lower the cost of commercial adsorbents. The chitosan derived from chitin is a naturally available bio-sorbent that could purify wastewater containing dyes in aqueous solutions [58].

### 6.5. Animal Feed

Research into BSFL frass for animal feedstock by Yildirim-Aksoy et al. [59] has shown excellent results in improving the growth of hybrid tilapia and enhancing the resistance of innate immune components and resistance to bacterial infection. This study fed five diets containing varying percentages of BSFL frass of 0, 5, 10, 20, and 30% to juvenile hybrid tilapia as a partial substitute for a mixture of soybean meal, wheat short, and corn meal on an equivalent protein basis. The improvement in hybrid tilapia growth could be due to the high protein content of BSFL frass. The fish diet containing 5 to 30% BSFL frass has a slightly higher protein efficiency than the diet without BSFL frass fed to the control group. The result showed that BSFL frass could be used as a partial replacement for commercial animal feed.

### 6.6. Feedstock

If necessary, BSFL frass can be fed to a biogas plant for anaerobic digestion for further processing. This post-processing of BSFL frass could simultaneously resolve problem with the handling of wet, high moisture content BSFL frass and to reuse the potential energy sources from wet BSFL frass to run the plant [5,12]. Post-processing by anaerobic digestion or composting could prevent soil nitrogen shortages or decreases in soil gas permeability as it increases the porosity and stability of wet BSFL frass [23]. Meanwhile, a preliminary study by Newton et al. [60] shows that BSFL frass from swine manure processing is a suitable feedstock for vermiculture. Processing the BSFL frass with earthworms will help in producing an additional product (earthworms) and transforms the residue into a product with proven economic value for the horticultural industry, earthworm castings.

## 7. Challenges of BSFL Frass on a Food Waste Substrate

### 7.1. High Moisture Content

Even though the BSFL treatment process may be sufficient for producing BSF larvae, the BSFL frass might be immature or unstable compost product because of its high moisture content. The characteristics of BSFL frass are influenced primarily by the properties of the substrate fed to BSFL [33]. Many organic waste substrates have a high moisture content (>80%) [61]. Food waste from human consumption has a very high moisture content (about 85%), which is a favourable condition for BSFL production and could rapidly degrade food waste [3]. However, the BSFL product from the food waste substrate has produced moist, grey, and clay-like texture. Although this wet BSFL frass does not represent mature compost characteristics, it also contains high ammonium concentration and has low porosity, that could stunt plant growth when being applied as a soil amendment [27]. Kawasaki et al. [36] also has conducted an in-depth investigation on the agriculture value of BSFL frass from food waste substrate. The result showed that the BSFL frass has a higher ammonium nitrogen concentration but lower nitrate–nitrogen content showing that BSFL frass in an anaerobic condition due to the present of high moisture content. Nitrate serves primarily as a source of nitrogen that ensures sufficient nutrition for plant development and soil microorganisms [62]. A substrate with a high moisture content could also reduce the efficiency of a BSFL treatment process and dry separation of BSF larvae and frass [61]. Cheng et al. [63] reported that dry separation of the larvae from the frass is not possible when the moisture content of a food waste substrate exceeds 80%. However, the BSFL frass can be easily separated from the insect biomass using a 2.36 mm sieve when the moisture content of the food waste is 70 to 75%. Reducing moisture content in the BSFL treatment process could result in slower BSFL growth. The BSFL also tend to crawl out of the treatment container as high moisture content will lead to a lower temperature for BSFL to live at as they need a mesophilic temperature (~30) for optimum waste conversion [31]. The wet separation process of BSFL frass is also cumbersome and time-consuming if the moisture content has not evaporated sufficiently during the BSFL composting [39,61,63]. The BSFL frass must have 50% dry matter content for easy separation of the BSFL frass from the larvae [63]. The beneficial effect of BSFL frass with low moisture content is good for soil aeration and solubility; on the other hand, the BSFL frass with a high moisture content could have inadequate oxygen supply for the plant [23]. The adverse impact of BSFL frass leachates caused by the excess moisture content could also cause ammonia poisoning in the plant and stunt plant development if the BSFL frass is not appropriately applied [18].

### 7.2. Ecological Risk

Biowaste characteristics, including biowaste and BSFL gut microbes, determine the properties of the BSFL frass [7]. When food waste contaminated with mercury was used as a substrate, the BSFL frass harvested after 13 days of composting treatment had a mercury level below the European Union’s (EU’s) threshold value of 0.7–10 mg Hg/kg [20]. BSFL frass macronutrient analysis showed a high nutrient content which is necessary to support plant development. Even though BSFL reduces the heavy metals content in BSFL frass, the ecological risk posed by the BSFL producing BSFL frass containing pathogenic microorganisms is a grave concern [33]. The type of substrate determines the BSFL gut microbiome, and the BSFL excrement determines the microbiome in the BSFL frass [23]. Previous studies have shown that BSFL can reduce *Escherichia coli* [32,36] and *Salmonella* spp. [55] in the initial substrate; however, some studies have identified the presence of potential foodborne pathogens, such as *Salmonella* spp. and *Bacillus cereus* [64]. Kawasaki et al. [36] found a low presence of *Escherichia coli* in BSFL frass, but a disease-causing bacterium in plants, *Xanthomonadaceae*, is present in BSFL frass harvested from the treatment of food waste substrate. BSFL frass harvested from food waste (fruit/vegetable mix waste) serves as a reservoir for coliforms and gram-negative bacteria. The negative microbial community in BSFL frass could come from the microbial community of the initial substrate, and the inactivation of it by sterilization using high-energy electronic beam is rather unpromising result for BSFL rearing. Gold et al. [53] reported that inactivating the microbial community in the initial food waste substrate reduced the efficiency of BSFL rearing, indicating that the microbial community of the initial food waste substrate is beneficial for substrate decomposition and/or BSFL growth. It is essential to conduct more research on the ability of common species of the BSFL intestinal microbiota (*Providencia*, *Dysgonomonas*, *Morganella,* and *Proteus*) due to their abundance in the BSFL frass produced from food waste substrate. Furthermore, both pre and post treatment could be crucial to increasing the revenue from BSFL frass as it may give significant impact to the maturity and stability of BSFL frass.

## 8. Pre-Treatment of the Substrate

### 8.1. Substrate Particle Size

It is crucial to ensure that the initial substrate is free from hazardous materials and inorganic substances before reducing the substrate particle size to 1–2 cm diameter because BSFL cannot break down large substrate particles with their mouthparts. Furthermore, increasing the substrate surface area could also promote the growth of beneficial microbial communities in the substrate [39]. Gao et al. [25] mentioned that shredded maize straw into small pieces by using a straw crusher helps for easy digestion to BSFL.

### 8.2. Substrate Moisture Content

Dortmans et al. [39] suggested controlling the substrate moisture content by grinding it in a kitchen blender to reduce its particle size and obtain a homogeneous substrate for easy BSFL digestion. The slurry substrate (>80% moisture content) must be dewatered or mixed with another type of waste or more dry waste. There are various ways to dewater the substrate or waste. The low-cost method uses a cloth bag to filter the excess water [39] or perforated containers to drain the water by gravity [13]. By allowing moisture content of substrate greater than 80%, active ventilation may be added for attaining dry separation of BSFL prepupae and dry BSFL frass [61]. Food waste substrate must have a moisture content of 70 to 75% to allow dry separation of the BSF larvae from the frass [63]. However, water must be added to a substrate with a moisture content of less than 70% [39]. Alternatively, dry waste, such as rice bran, palm kernel expeller, and soybean curd residue can be added to control organic waste with high moisture content [65,66]. As studied by Y. Xiao et al. [66], the addition of 15% rice bran to chicken manure and pig manure observed to be better in BSFL development and higher conversion rate. The addition of rice bran to the chicken manure and pig manure has improved BSFL development while lowering viscosity and improving organic waste ventilation. The treatment of organic waste by BSFL in an environment with good ventilation could produce BSFL frass with optimal moisture content.

### 8.3. Addition of Effective Microorganisms (EM)

Z. Liu et al. [67] discovered that lignin inhibited larval growth considerably, indicating that it is essential to add lignin-digesting microbes to the lignin-rich substrates fed to BSFL. The result showed that high lignocellulosic substrate must be pre-treated by adding effective microbes to prevent stunted BSFL growth. BSF larvae must access the epidermal layer of the plants before consuming the often difficult-to-digest complex organic compounds, such as lignocelluloses [65]. Gao et al. [25] reported that fermentation improves the digestibility and palatability of maize straw and, thus, extends the larval stage of BSFL since the larvae continue eating the foodstuff. Isibika et al. [68] explored several pre-treatment methods, including microbial, chemical (non-protein nitrogen), heat-based, and combinations of the three pre-treatment methods, to treat banana peel. They found that the ideal pre-treatment method is adding effective microorganisms to achieve maximum substrate degradation and facilitate larval uptake of the released nutrients. There are two methods for adding effective microorganisms to the substrate or microbial fermentation of the substrate, in situ fermentation, and ex situ fermentation. In in situ fermentation, the effective microorganisms spiked simultaneously into the substrate in the BSFL treatment process; in ex situ fermentation, the microorganism ferments the substrate before the BSFL feeds on it [65].

Because a single substrate might not have sufficient nutrients for BSFL development, in situ, and ex situ fermentation in the presence of exo-microbes (for example, *Saccharomyces cerevisiae*, *Bacillus subtilis*, *Lactobacillus buchneri*) could enrich the feeding substrate. Once the BSFL has reached maturity, its biomass can be collected and used as a feedstock for proteins, lipids, and other biochemical syntheses [65]. The hydrolysis process via microbial fermentation breaks down the complex components of organic matter while the BSFL populations in the wild decompose decaying organic matter [65,69]. According to Raksasat et al. [65], the benefit of in-situ fermentation is that it increases BSFL growth rate, survival rate, and waste-to-biomass conversion in wastes rich in fibre and lignocellulosic wastes, for greater digestion and nutrient assimilation into BSFL bodies. In situ bacteria may also aid in the absorption process by providing vital nutrients for BSFL growth and protecting the growing BSFL from threats such as parasitoids or diseases. However, ex situ fermentation is most preferable for recovery of protein-rich wastes, such as kitchen waste as it may enhance the nutritional properties of lignocellulosic biomass and providing sufficient digestible food waste composition, such as protein and fat and, therefore, increasing yield of BSFL lipid and protein content [70].

Gold et al. [53] investigated the role of microbial population in the BSFL treatment process and evaluated the influence of non-sterile and sterile food waste on BSFL rearing performance. They concluded that inactivating the microbes in the substrate has reduced the rearing performance, indicating that the initial substrate microbial communities contribute to substrate degradation and larval growth. The poor rearing performances has, therefore, reduced the microbial populations and altered the BSFL frass’s physicochemical properties and composition. The BSFL reared on sterile food waste substrate also has produced BSFL frass with a rich microbiota, *Providencia*, *Dysgonomonas*, and *Morganella*, transferred from the BSFL intestine which need a further study on its role in affecting rearing performance. This implies that it is not necessary to sterilize food waste that will be used as substrates as the initial food waste microbiota, dominated by lactic acid bacteria could aid in BSFL composting and increasing rearing performance.

### 8.4. Addition of Protein-Rich Substrate

A single waste substrate containing lack of nutrients, such as protein and fat, could hinder the BSFL growth and reduce the BSFL bioconversion efficiency in producing high-quality by-products. Protein is a crucial component of the larval feeding substrates and could affect the BSFL’s ability to complete its life cycle [53]. Lignocellulosic wastes made up of lignin, cellulose and hemicellulose could stifle BSFL digestion. Therefore, adding or blending the main substrate with rich-nutrient and low-cost discarded organic waste (for example, palm kernel expeller and soybean curd residue) could improve nutritional balance, such as the C/N ratios and pH value of the substrate, and improve BSFL rearing efficiency [65]. If there is difficulty in obtaining discarded organic waste, commercial chicken feeds are a suitable addition to a protein-rich substrate. Spranghers et al. [69] has affirmed that commercial chicken feed is a high-quality reference substrate because it is a feedstock formulated for broilers and, thus, has a high protein content. Chicken feed mixed with water (60–70% moisture content) was used as an attractant and rearing substrate for house flies, and studies have shown that it is also suitable for rearing BSFL [71,72]. Sprangers et al. [69] found that larvae reared on chicken feed have the highest development rate compared to vegetable waste, biogas digestate, and restaurant waste. The first prepupae in the chicken feed substrate emerged after 12 days, while the first prepupae in vegetable waste and biogas digestate substrates emerged after 15 days. The high amount of oil or grease in the restaurant waste substrate hamper BSFL development, and the first prepupae emerged after 18 days. They take longer to mature because BSFL cannot digest oil or grease efficiently [73].

## 9. Post-Treatment of BSFL Frass

Researchers have suggested that post-processing of BSFL frass into vermicomposting facility and biogas plant is important because of its maturity due to rapid composting by BSFL [12,22,39,60,74]. However, post-treatment of BSFL frass using the composting method is the easiest and most suitable for generating revenue from the BSFL treatment. According to Dortmans et al. [39], wet BSFL frass can be composted with other garden waste or any waste with carbon sources available. Carbon source products from the BSFL facility, such as chitin (pupae exuvia, dead imago), can be added for BSFL frass composting [33].

The study of Beesigamukama et al. [38] has stated that harvested BSFL frass after two weeks of bioconversion and composted the BSFL frass using the heap methods for five weeks to obtain mature and stable frass compost. The composted BSFL frass was used as a fertiliser to determine its effect on maize production. The maize plots treated with BSFL frass produced bigger plants with the highest chlorophyll concentration and grain yields than the commercially treated maize plots. The maize treated with composted BSFL frass at 2.5 t ha^−1^ and 30 kg N ha^−1^ has higher nitrogen recovery efficiencies than commercial fertilisers. Song et al. [37] found BSFL frass to be a sustainable compost more than a fresh BSFL frass. As reported by Song et al. [37], immature BSFL frass was harvested from rapid composting when further composting of BSFL frass by using forced aeration method has yield larger pak choi plant than those grown in fresh frass.

Dry BSFL frass should also undergo further treatment as a maturation process because of the rapid composting of organic waste. The maturation process is necessary to reduce the microbial activities in the BSFL frass before applying them as a soil amendment or organic fertiliser since they must take up oxygen and nitrogen in the soil or plants [39]. BSFL frass may undergo further decomposition on a concrete floor under a roof for three weeks before using it as an organic fertilizer [17]. Alattar et al. [27] suggested that it is better to pre-dry the BSFL frass so that it can be used as a fertiliser in a ratio of one part frass to two parts soil.

## 10. Direction of Future Research

Researchers referred to BSFL frass as BSFL digestate, as well as the possibility of BSFL digesting uneaten feed. The term BSFL frass can also refer to a combination of BSFL digestate, uneaten substrates, and BSFL chitin supply. The chitin source could be dead BSF larvae, BSF shedding, or dead fly [33]. To the authors’ knowledge, there is a chance to study the properties of the frass mixtures containing the additional chitin source to explore its potential when it is widely used in commercial applications, especially on organic fertilisers [75].

Even though organic fertilisers can be produced from BSFL frass-based fertiliser, future studies should determine its nutrient supplementation, depending on the requirement of the different types of crops since the composition of BSFL frass varies greatly, and the nutrient content (NPK) is significantly dependent on the feed substrate [40]. Future research should also investigate the impact of pre-treating food waste on the maturity and stability of BSFL frass. The recommended pre-treatment for food waste is adding a suitable low-cost organic waste with protein-rich ingredients to control the moisture content of the food waste. Food waste as a single substrate also has proven immaturity of BSFL frass mainly in tropical region due to their high moisture content BSFL frass generated greater than 45%, producing frass in wet, slurry, and in anaerobic conditions. There is a need to study the substrate composition that could enhance nutrient content of BSFL frass (NPK) and improves BSFL frass end moisture content to reach maturity level (30–45%). According to Jalil et al. [76], BSFL prefers protein meal waste over carbohydrate food waste, indicating the importance of protein for their growth. Inconsistent nutrient content, such as protein in the food waste are the primary factors affecting the development, production, and efficiency of the BSFL frass. In Malaysia, a study of composition of food waste has been performed by Chua et al. [77]; food waste collected from university’s café, restaurants, markets, and household recorded more than 50% carbohydrates content compared to less than 35% of protein content in food waste. Pre-treating the food waste with effective microorganisms and fermentation may help in promoting the maturity of decomposition of food waste. The presence of beneficial exo-microbes, such as *Saccharomyces cerevisiae*, *Bacillus subtilis*, *Lactobacillus buchneri*., may improve the digestibility and palatability of the substrate, thus making it suitable for use as an organic fertiliser [25].

This review paper has discussed post-treatment of BSFL frass via simple composting, feedstock for vermicomposting and feedstock for the biogas plant. However, some recent studies discovered the environmental impact of the post-treatment of BSFL frass due to the emission of greenhouse gas. Therefore, it is essential to identify the composting technique that reduces greenhouse gas emissions. A study by Song et al. [37] identified three types of frasses that have a lower global warming potential than incineration; the fresh frass has the lowest emission, the frass composted via forced aeration has the second-lowest emissions, and the simple composted frass has the highest emission. Thus, considering the compost quality and environmental impact, the most practical post-treatment option for additional composting of BSFL frass is forced aeration. However, Lopes et al. [40] opined that using BSFL frass as a feedstock for earthworms in vermicomposting helps stabilise and mature the frass while providing a high-value output as goods. Vermicomposting on a smaller scale could counter the problem of greenhouse gas emissions in the BSFL post-treatment via the conventional method.

## 11. Conclusions

BSFL frass has many unique properties that are good for the environment, particularly soil and plant nutrition cycling from food waste substrate. In addition to soil amendments and organic fertiliser, BSFL frass has promising potential for many applications, including animal feed, growth media, biochar, and feedstock for vermicomposting and biogas plants. Although the qualities of food waste substrate could affect BSFL frass quality, pre-treatment of food waste, and post-treatment of BSFL frass could produce high-quality frass.

## Figures and Tables

**Figure 1 foods-11-02664-f001:**
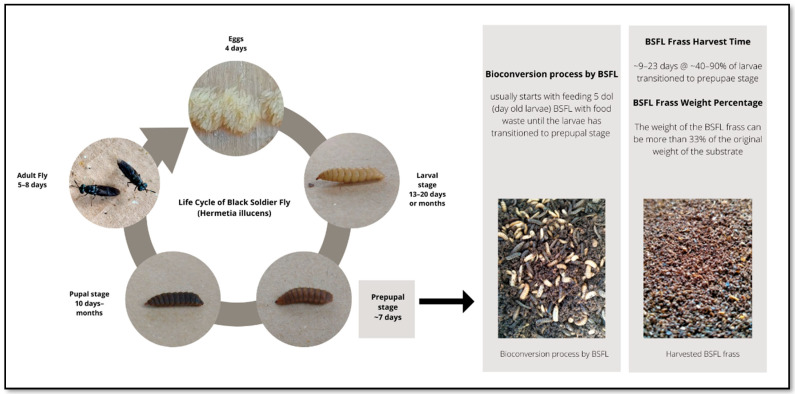
Life cycle of black soldier fly and the production of BSFL frass.

**Figure 2 foods-11-02664-f002:**
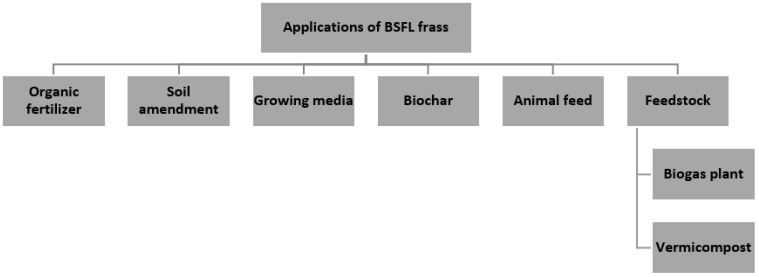
Applications of BSFL frass.

**Table 1 foods-11-02664-t001:** The physiochemical characteristics of BSFL frass from different food waste types.

Type of Food Waste	pH	C/N Ratio	Moisture (%)	Temperature (°C)	Total Nitrogen as N (%)	Total Phosphorus as P_2_O_5_ (%)	Total Potassium as K_2_O (%)	References
Kitchen waste	7.0	8:1	50	-	-	-	-	[35]
Kitchen waste	7.4	17:1	63	-	-	-	-	[3]
Municipal food waste	7.3–7.4	8:1–9:1	63–65	26.3–26.5	-	-	-	[30]
Household food waste	7.4	17:1	56	-	2.2	0.1	0.1	[36]
Food waste, chicken faeces, and sawdust (3:2:1 ratio)	6.1–8.0	-	72	27.0	1.7	1.1	2.1	[20]
Fruits and vegetables	5.6	27:1	10	-	1.8	-	-	[23]
Maize straw	8.0	-	38	-	0.6	2.5	2.1	[25]
Okara and wheat bran	7.5	8:1	-	-	4.8	1.0	0.9	[37]
Okara and wheat bran	7.7	10:1	-	-	3.2	0.8	0.5	[37] *
Brewery spent grain	7.7	17:1	30	-	2.1	1.2	0.2	[38]

* Frass obtained from post-treatment of thermophilic composting. References without an asterisk indicate referred study was using fresh harvested frass.

**Table 2 foods-11-02664-t002:** The physiochemical characteristics of different types of fertilisers.

Type of Fertilisers	Type of Food Waste	pH	C/N Ratio	Moisture (%)	Temperature (°C)	Total Nitrogen as N (%)	Total Phosphorus as P_2_O_5_ (%)	Total Potassium as K_2_O (%)	References
Compost from windrow composting	Food waste and yard waste	6.8–7.4	17:1–23:1	46–61	26–28	-	-	-	[43]
Compost from composting bin	Cafeteria food waste and yard waste	7.5	-	-	30	0.9	0.8	0.4	[44]
Compost from aerated composting	Household food waste	6.5–7.5	36:1	32–33	32	0.9–1.0	0.6–0.7	0.9–1.0	[45]
Digestate from anaerobic digester	Municipal food waste	8.1	11:1	85	-	9.6	2.4	2.3	[46]
Commercial organic fertiliser		-	-	-	-	2.3	2.3	2.3	[44]
Commercial chemical fertiliser		-	-	-	-	16	16	16	[44]
Preferred for agronomic purposes		6–8	<30:1	30–45	Ambient temperature	>0.6	>0.22	>0.25	[30,37,44,47,48,49]

**Table 3 foods-11-02664-t003:** The composition of food waste attribute to BSFL frass composition.

Sample	Type of Waste	Nutrient Composition ^e^					BSFL Frass Composition			
		Protein	Fat	Calcium	Starch	Sugar	NH_4_-N ^f^	PO_4_-P ^f^	K ^e^	References
1	Pollinator hulls ^a,b^	40.1	31.0	2.4	5.33	152.7	615.0	355.0	35.4	[50]
2	Nonpareil Hulls ^b,c^	46.3	20.5	2.1	3.57	243.5	5695.0	227.5	38.5	[50]
3	Pollinator Hulls ^b,c^	41.0	24.8	2.3	5.03	178.1	755.0	405.0	44.6	[50]
4	Nonpareil Hulls ^c,d^	55.3	22.3	2.2	4.23	291.3	5052.5	515.0	44.6	[50]
5	Monterey Hulls ^c,d^	67.7	26.5	2.8	4.33	119.2	400	825.0	43.4	[50]
6	Pollinator Hulls ^c,d^	40.6	22.9	2.6	4.93	202.1	1420.0	485.0	36.0	[50]
7	Mixed Almond Shells ^c,d^	42.6	14.6	1.9	3.65	53.2	2595.0	245.0	17.9	[50]

^a^ harvest year: 2016, ^b^ product origin: Chico region, ^c^ harvest year: 2017, ^d^ product origin: Buttonwillow region, ^e^ quantity in g kg^−1^ dry matter, ^f^ quantity in ppm.

**Table 4 foods-11-02664-t004:** Benefits of BSFL frass.

Benefits of BSFL Frass	Reference
BSFL frass contains chitin that improves the soil microbiome	[15]
Rich in nutrients (macronutrients, micronutrients, and organic matter)	[19,20,25,30]
High phosphorus concentrations in the BSFL frass promote nitrogen accumulation in plants	[23]
BSFL frass contains chitin that naturally produces antimicrobial peptides, which serve as a defence barrier for the plant	[34]
Beneficial microorganism population for plant uptake	[34,53]

## Data Availability

Data is contained within the article.

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
