# Peer review of "Potential Applications of Frass Derived from Black Soldier Fly Larvae Treatment of Food Waste: A Review"

_foods, 2022, doi:10.3390/foods11172664_

Round 1

Reviewer 1 Report

I have reviewed the paper ‘potential applications of frass derived from black soldier fly larvae treatment of food waste: a review’. Overall, this paper is well written and easy to follow. It also provides useful information to the current hot topic ‘food waste’ area. Some of my comments:

1.     Line 21: please delete ‘maybe’

2.     Line 34-35: please add reference

3.     Figure 1: please identify dol

4.     Line 88: what about the beginning year of the data base

5.     Table 1: this table is too simple, the author could consider add other parameter comparisons such as pH, temperature you mentioned in the manuscript herein to give a stronger and more informative table

6.     The authors touched a little bit on the types of food waste on BSFL frass composition, however, it will be more interesting that the author provide more detailed information (one section) discussing how the composition such as sugar, protein, fiber, fat, moisture, etc. will affect the frass composition, especially considering the composition of different types of food waste varies a lot. This information will be extremely helpful to the whole area.

7.     It is suggested the author provide a table lists different types of food waste, composting conditions such as pH, moisture content, C/N, temperature, and frass yield. This could provide a clearer overview of current frass generation from food waste related research

8.     How about the economic and environmental perspectives of this approach (generating frass from food waste), are there any research papers talk about these two areas?

Reviewer 2 Report

Potential Applications of Frass Derived from Black Soldier Fly 2 Larvae Treatment of Food Waste: A Review 3

This is a review of BSFL frass produced from food wastes. The review describes the properties of the BSFL frass, benefits and applications. The authors also report on the pre-treatment options of the food wastes and post-treatment of the BSFL frass.

Generally, the manuscript is well written. To improve the manuscript, the authors could benefit from the comments in the document. The authors should also work on several typo errors including direct and indirect citation errors in the text.

Line 19 - frass, is not known to be a beneficial product?  What do you mean?

The author could rewrite the sentence -“as many aspects related to frass are still unknown”. Furthermore, The value of BSFL frass is quite known.  Rephrase to "limited information is available". Examples are papers by Beesigamukama et al., 2020 and 2021.

Line 20 - Rephrase " The applications of BSFL frass can be intensified by understanding...."

Line 22- “The generated BSFL frass could be exceed 33%” change to “the generated BSFL frass could exceed 33%

Line 26 – delete issue And insert challenges

Line 72- “to be reuse as a fertilizer” can change this to “to reuse as a fertilizer” or “to be reused as fertilizer”

Line 74- change “high value” to “high-value”

Line 75- “Revenue loss” could be changed to “revenue losses”

Line 95- check on the referencing styles

Line 111- “Dry BSFL frass produced mainly” change to “Dry BSFL frass is produced mainly”

Line 112- “Meanwhile wet BSFL frass, can be seen as a dense” could be written as “Meanwhile, wet BSFL frass can be seen as a dense, grey in colour…..”

Line 125 - Can it also be concluded that the weight of BSFL frass depends on the substrate?

Line 154 – Please explain what is immature compost? Please explain the importance of the rapid composting of the BSFL frass

Line 155 - Please elaborate further on the maturity ans stability of BSFL frass. What are the properties of a mature and stable frass?

Table 1: indicate the source of food waste. That is BSFL frass from may be kitchen waste

Line 180- The author could include a comparison of nutrition contents reported by different studies using different substrates

Line 181 - The authors could give information whether there are specific food wastes with higher nutrient content than others.  They can then recommend such food wastes for the production of BSFL frass fertiliser with high nutrient content

Table 2: BSFL frass was produced from which food waste?

Line 211 - Does this statement also imply that BSFL frass fertiliser is also at risk of being contaminated with these heavy metals?

Line 215 - The authors can also include the economical and ecological benefits of BSFL frass fertiliser

Line 225- “BSFL frass has aid in promoting nitrogen” change to “BSFL  frass has aided in promoting nitrogen”

Line 244 - Given that the paper is focusing on food wastes, the authors should give more details on the BSFL frass fertiliser produced from these food wastes.  That is properties of frass from specific wastes and highlight the best food waste substrates

Line 265- 266- rewrite the sentence in past tense. Also check the tenses in line276 and anywhere else in the main text.

Line 277- change plant to “plants”

Line 285- “a high toxic dye” could change to “a highly toxic dye”. Also rewrite the sentence “a cationic dye which is a high toxic dye that may contain in wastewater” replace “that may contain in wastewater” with “that may be found”

Line 310-  “Meanwhile, preliminary study” could change to “Meanwhile,  a preliminary study”

Line 312- “Processing the BSFL frass with earthworms will help in produces” change to “Processing the BSFL frass with earthworms will help in producing…”

Line 326- “it also content high ammonium” replace “content” with “contains”

Line 425 - Among the two fermentation methods, which is the most effective that can be recommended for pre-treatment of food wastes?

Line 441 - Does this statement then imply that it is not necessary to sterilize food wastes that will be used as substrates for BSFL?

Line 444 - Please explain what will hinder BSFL growth, given that the single waste substrate has protein and fat

Line 467 - Please explain further on this maturity in terms of BSFL frass properties

Line 477 - Since the review is about the BSFL frass, the authors should give more details or properties of a mature and stable frass compost. What are the differences between immature and mature frass compost?

Line 509 - These words are not clear. What makes the BSFL frass immature? moisture content, nutrient content?

Line 512 - maturity in relation to nutrient content?

Line 520 - maturity in terms of the level of composting?

Line 541 - The review could also conclude by recommending specific food wastes that can be used for the production of high quality BSFL frass fertilizer in terms of nutrient content and other physicochemical properties

Round 2

Reviewer 1 Report

All comments ara well addressed and I recommend acceptance of the paper.